# MAGIC: Multi-domain Analysis and Generalization of Image manipulation localization

## Abstract

Advanced image editing software enables easy creation of highly convincing image manipulations, which has been made even more accessible in recent years due to advances in generative AI. Manipulated images, while often harmless, could spread misinformation, create false narratives, and influence people's opinions on important issues. Despite this growing threat, current research on detecting advanced manipulations across different visual domains, remains limited. Thus, we introduce Multi-domain Analysis and Generalization of Image manipulation loCalization (MAGIC), a comprehensive benchmark designed for studying generalization across several axes in image manipulation detection. MAGIC comprises over 192K images from *two distinct sources* (user and news photos), spanning a diverse range of topics and manipulation sizes. We focus on images manipulated using recent *diffusion-based* inpainting methods, which are largely absent in existing datasets. We conduct experiments under different types of *domain shift* to evaluate robustness of existing image manipulation detection methods. Our goal is to drive further research in this area by offering new insights that would help develop more reliable and generalizable image manipulation detection methods. We will release the dataset after this work is published.

## 1 Introduction

With the advent of sophisticated image editing tools like Photoshop and the emergence of advanced generative AI models (Goodfellow et al., 2014; Sohl-Dickstein et al., 2015; Rezende & Mohamed, 2015), image manipulations have achieved an unprecedented level of realism (Shi et al., 2020; Saharia et al., 2022; Yu et al., 2023), making it challenging to distinguish between genuine and altered content. Modern image manipulation techniques can produce convincingly fabricated materials that could be used to create false narratives, misrepresent key individuals (Guo et al., 2023), and spread misinformation. Image manipulation datasets play an important role in countering such fabrications by providing the essential resources needed for training manipulation detection models that are designed to localize altered image regions (Liu et al., 2022; Wang et al., 2022; Guo et al., 2023). However, as shown in Figure 1, prior datasets focus on one axis of generalization, namely across manipulation types. Moreover, prior works typically train and test on the same data domain. We argue that it is important to study generalization across multiple axes, such as training and testing models across different image sources, semantic topics, manipulation types, sizes, etc.

To this end, we introduce Multi-domain Analysis and Generalization of Image manipulation loCalization (**MAGIC**), a carefully curated, large-scale image manipulation dataset. Our dataset is created from two image sources, VisualNews (Liu et al., 2020) and MS COCO (Lin et al., 2014). VisualNews includes images from four major news outlets—The Guardian, BBC, USA TODAY, and The Washington Post. To ensure image diversity, we sample images from eight different categories, such as Business or Science. To explore significant domain shifts, we also sample images from MS COCO, which contains user photos obtained from Flickr. Most existing datasets focus on traditional manipulation techniques such as splicing and copy-move, and do not include any deep learning-based inpainted images, with an exception of Tânțaru et al. (2024) (see Table 1). In contrast, we utilize various open-sourced diffusion-based inpainting techniques, including Blended-Diffusion (Avrahami et al., 2022), Stable-Diffusion (Rombach et al., 2022b), Latent-Diffusion (Rombach et al., 2022b; Blattmann et al., 2022), GLIDE (Nichol et al., 2022), Blended-Latent Diffusion (Avrahami et al., 2023) and GLIGEN (Li et al., 2023) to add or remove objects from images. We also include a widely

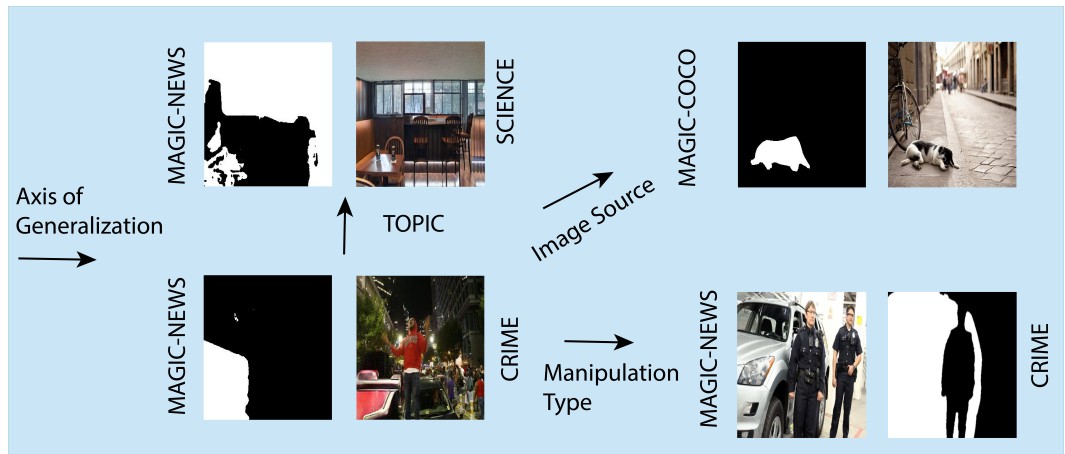

Figure 1: We illustrate the axes of generalization within MAGIC, a comprehensive benchmark for image manipulation detection. Typically, manipulation detection datasets focus on one axis of generalization, namely unseen manipulation types (e.g., Splicing vs. CopyMove). For MAGIC however, we have three such axes: image source (MS COCO vs. VisualNews), topic source (e.g., Science vs. Crime) and manipulation type (e.g., Blended Diffusion vs. Stable Diffusion), which could be extended to include other axes of generalization.

Table 1: We provide a comparison of our dataset, MAGIC, to several prior datasets w.r.t.: overall dataset size (Images), number of the authentic (AU) samples, traditional manipulation techniques (Trad. MT), e.g., splicing, copy-move, and morphing, and advanced methods, i.e., GAN-based inpainting (GAN-INP) and diffusion-based inpainting (Diff-INP). The datasets include COLUMBIA (Hsu & Chang, 2006), CASIAV1 and CASIAV2 (Dong et al., 2013), COVERAGE (Wen et al., 2016), DEFACTO (MAHFOUDI et al., 2019), and DOLOS (Tânțaru et al., 2024). MAGIC stands out with its size, emphasis on diffusion-based inpainting, and inclusion of two image sources (MS COCO and News), surpassing the scope of previous datasets by incorporating three axes of generalization.

| Dataset | Images | Trad. MT | GAN-INP | Diff-INP | AU | Domain |
|---|---|---|---|---|---|---|
| COLUMBIA | 1,845 | 912 | | | 933 | CalPhotos |
| CASIAV1 | 1,721 | 921 | | | 800 | Corel Dataset |
| CASIAV2 | 12,323 | 5123 | | | 7,200 | Corel Dataset |
| COVERAGE | 200 | 100 | | | 100 | In/Outdoor Scenes |
| DEFACTO | 229,000 | 229,000 | | | | MS COCO |
| DOLOS | 148,112 | 10,800 | 10,800 | 105,812 | 20,700 | CelebA, FFHQ |
| MAGIC (ours) | 192,597 | | | 162,933 | 29,664 | MS COCO, News |

used proprietary tool, Adobe Firefly (Adobe, 2024). We refer to our data subsets as **MAGIC-News** and **MAGIC-COCO**, respectively. Table 1 shows how existing datasets compare to ours; they lack in coverage of various realistic image domains, e.g., news images. Additionally, due to the lack of any context (to accompany the images), no distinctions based on different topics have been studied. Figure 2 provides several examples of our dataset, showcasing its diversity.

Our MAGIC dataset offers the following key benefits: (1) We ensure that it allows exploring significant domain shifts by leveraging two image sources and generating diverse manipulations.(2) By sampling images from eight topics within the VisualNews subset (e.g., Business, Sports) our dataset enables us to study whether topic distribution affects model performance. This is particularly relevant for news images, where the context can influence the interpretation of manipulations. (3) To evaluate the realism of our dataset, we conduct a human perceptual study, where participants assess image and object realism. We compare human perceptual scores with the predictions of the detection models to analyze any trends that occur . This empirical study highlights the challenges and effectiveness of current detection methods when applied to our dataset.

In summary, our work makes the following main contributions:

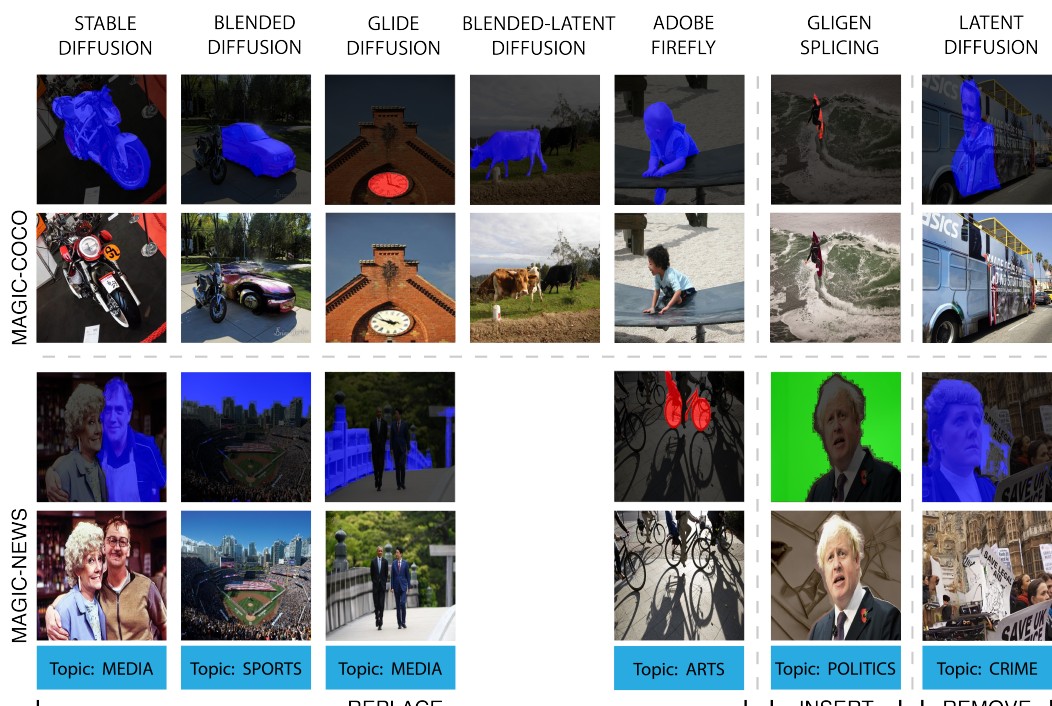

Figure 2: Examples from our MAGIC dataset, which contains 192,597 images from two visually distinct domains: MS COCO (Lin et al., 2014) and Visual News (Liu et al., 2020). It also spans 8 distinct topics within the news domain, 4 of which are depicted here. The dataset contains 7 state-of-the-art manipulation techniques classified into three categories: removal, replacement, and insertion. Manipulations cover a wide range of sizes, from 1% to 100% of the image area, with coverage indicated by color: red for small, blue for medium, and green for large size. These examples demonstrate the variable perceptual quality and diversity of manipulations in our dataset.

- We introduce MAGIC, a novel large-scale image manipulation detection dataset consisting of 192,597 image-mask pairs. Our dataset exhibits diverse distributions across various dimensions, including image sources and topics, manipulation types, and sizes.
- Our dataset is the first manipulation dataset that explores domain generalization across three axes, namely it contains 7 image manipulations across two different image sources, plus for the **MAGIC-News** we group images by topics. We reveal that current models struggle with out-of-distribution (OOD) samples. We conduct experiments that shed light on such unique challenges posed by our dataset that could provide valuable insights for future research directions.
- We showcase how recent transformer-based approaches for in-distribution (ID) detection tends to perform well compared to CNN-based approaches even on diffusion-based inpaintings. However when it came to large manipulation sizes most models tended to perform poorly especially when it came to OOD detection for models trained on **MAGIC-COCO** and tested on **MAGIC-News**.
- We explore the impact of domain generalization techniques applied to the top-performing manipulation detection models on our dataset. We aim to determine whether these techniques can improve robustness against image distribution shifts. We show that utilizing popular domain generalization techniques can struggle with improving performance across image source and manipulation type.
- We conduct a human perceptual study on a subset of our dataset. The results of this study are analyzed and compared to the predictions of various manipulation detection models.

## 2 RELATED WORK

**Datasets for Image Manipulation Detection.** In response to the growing complexity of image manipulation techniques, contemporary datasets have emerged, broadening the scope of research in this field. DEFACTO (MAHFOUDI et al., 2019) encompasses a substantial collection comprising 25K instances of inpainting, 19K cases of copy-move, 105K instances of splicing, and 80K instances

of morphing manipulations. GRE (Sun et al., 2024) also studies the effect of a complementary set of manipulation methods, but they did not consider the effects ond detection performance of shifts due to image sources, topics, or the quality of manipulations in their paper. DOLOS (Țânțaru et al., 2024), contains 148,112 images which includes around 105,812 diffusion based inpainted images. This dataset is mainly made up of human faces and utilizes FFHQ (Karras et al., 2019) and CelebA (Liu et al., 2015), where inpainting models like Repaint-P2, Repaint-LDM (Rombach et al., 2022a) were used for manipulations. Additionally, DOLOS includes 512 images inpainted using GLIDE (Nichol et al., 2022) with MS COCO (Lin et al., 2014) as the image source.

**Models for Image Manipulation Detection.** Among many recent manipulation detection models we have chosen four that have been shown to perform well on classical manipulation detection methods like copy-move, splicing etc. First, we have **PSCC-Net** (Liu et al., 2022) which employs a two-path process: a top-down path extracting local and global features, and a bottom-up,path for detecting manipulations and estimating manipulation masks across multiple scales. Next we have **HiFi** (Guo et al., 2023) which has a similar architecture to PSCC-Net but additionally employs a hierarchical fine-grained approach to classify forgery attributes at different levels, from general to specific. Next is **DOLOS** (Țânțaru et al., 2024), this model uses Patch–Forensics (Chai et al., 2020), which is a truncated image classification network that takes the feature activations after a few layers and projects them to a patch-level score using $1 \times 1$ convolutions. Finally we utilize **EVP** (Liu et al., 2023), which is a vision transformer based approach that use a frozen backbone SegFormer (Xie et al., 2021) and only contains a small number of tunable parameters to learn task-specific knowledge from the features of each individual image itself. We study two domain generalization techniques, **SWAD** (Cha et al., 2021), that improves robustness by averaging model weights during training time and seeking flat minima, and **Model Soups** (Wortsman et al., 2022), that leverages different versions of the same model with different hyperparameters and uses a greedy approach to average their weights to improve robustness. We apply both methods to our best performing model, EVP.

## 3 MULTI-DOMAIN ANALYSIS AND GENERALIZATION OF IMAGE MANIPULATION LOCALIZATION (MAGIC)

Our MAGIC dataset comprises of two distinct subsets. MAGIC-News contains 90,481 images, with 75,899 manipulated images and 14,582 pristine images sampled from the VisualNews (Liu et al., 2020) dataset. MAGIC-COCO contains 102,116 images, with 87,034 manipulated images and 15,082 pristine images sampled from MS COCO (Lin et al., 2014). All the manipulated images are accompanied by ground-truth masks . We design our benchmark such that we can study how different axes of domain generalization affect image manipulation localization performance.

### 3.1 IMAGE MANIPULATION TECHNIQUES

While constructing our dataset, we sought to generate a wide variety of image manipulations that would reflect real-world scenarios, offering a significant challenge for models tasked with localizing these alterations. To achieve this, we employed seven major diffusion-based manipulation techniques: five perform replacement, one performs removal, and one performs insertion. These methods are applied to both MAGIC-News and MAGIC-COCO, ensuring high diversity of manipulated images.

#### 3.1.1 REPLACEMENT AND REMOVAL BASED MANIPULATIONS

We employ five diffusion-based models that specialize in replacement-based manipulations. Each of these models offers specific strengths and capabilities. **Blended-Diffusion** (Avrahami et al., 2022) utilizes the CLIP (Radford et al., 2021) model for text guidance, combined with a denoising diffusion probabilistic model DDPM (Dhariwal & Nichol, 2021) to create highly realistic, localized edits based on user-provided text prompts. **Blended-Latent-Diffusion** (Avrahami et al., 2023) extends this by operating in the latent space, improving both speed and accuracy; it enables more realistic image inpainting. We also employ **Stable-Diffusion** (Rombach et al., 2022b), which operates in the latent space of pretrained autoencoders, making it more computationally efficient while retaining high visual fidelity. It is particularly effective in generating class-conditional images with high-resolution. **GLIDE** (Nichol et al., 2022) is another model which also uses CLIP guidance and excels at generating photorealistic content. Finally, **Adobe Firefly** (Adobe, 2024) is included as a proprietary model

for generating diverse image manipulations, simulating real-world scenarios where such tools are widely used. All these models are applied to both the MAGIC-News and MAGIC-COCO images. For MAGIC-News, segmentation masks are generated using Mask2Former (Cheng et al., 2022), whereas for MAGIC-COCO, the existing masks provided by the MS COCO dataset are used. All the models use both text and image data to guide the generation of manipulated images. We identify an object randomly selected from the mask and replace that region with the new content based on the object class. For example, if the mask corresponds to a "cat", the information that the object is a cat is used as a prompt to guide the diffusion models in generating the new content (i.e., another cat).

Finally, we use **Latent-Diffusion** (Rombach et al., 2021), where instead of replacing the selected object, the entire region is removed (inpainted) to blend with the background.

### 3.1.2 Insertion-Based Manipulations

Insertion of new objects into an existing image is also known as *splicing*. The splicing process begins by utilizing Mask2Former (Cheng et al., 2022) to generate panoptic segmentation maps from images sourced from MS COCO and VisualNews. Panoptic segmentation provides a detailed and comprehensive breakdown of the scene by categorizing each pixel into specific objects (things) and background elements (stuff). This segmentation map $S$ serves as a blueprint, outlining where each object is located within the image. Once the segmentation map is obtained, **GLIGEN** (Li et al., 2023) is employed to generate a new object that fits contextually into the designated area. GLIGEN leverages the spatial and semantic information from the segmentation map to ensure that the newly generated object not only aligns with the surrounding elements in terms of position and scale but also blends seamlessly with the scene's overall aesthetics. The generated object is then carefully inserted into the original image $I_{\text{original}}$ using the object mask $M$ (Here the object mask $M$ is randomly chosen from the panoptic segmentation), resulting in a spliced image $I_{\text{spliced}}$, as described by:

$$I_{\text{spliced}} = \text{GLIGEN}(S) \odot M \oplus I_{\text{original}}$$

Where $S$ is the segmentation map, $M$ is the object mask, and $I_{\text{original}}$ is the original image.

### 3.2 Diversity in MAGIC Dataset

When constructing the MAGIC-dataset, we curate a wide range of scenarios to better capture the different challenges that an image manipulation detector may encounter in real-world applications. News imagery, in particular, covers an broad variety of content, as illustrated in Figure 2. To ensure diversity in image content, we select our images from a range of topics. The VisualNews dataset provides 159 different topic annotations. However, many of these topics are overlapping or closely related (e.g., *science technology*, *nanotechnology*, *technology*). To address this, we group similar topics using the clustering framework of Tan et al. (2022). This results in eight distinct categories (see Table 2). The VisualNews dataset draws from four major news outlets: USA Today, Washington Post, BBC, and The Guardian. We sample an approximately equal number of images from each news outlet and topic, ensuring balanced representation[1]. MS COCO does not have topic labels; for our MAGIC-COCO subset we sample 82 object categories, including the most common objects (*person*, *car*), and less frequent objects (*hairbrush*, *giraffe*). The combination of news-related imagery and everyday objects from MS COCO ensures that our dataset represents not just specialized journalistic content but also a broad spectrum of general, real-world scenes.

In the MAGIC-News we manipulate both the foreground (e.g., people, buildings) and the background elements (e.g., sky, terrain), resulting in a wide variety of manipulation sizes based on the panoptic segmentation masks predicted by Mask2Former. This is in contrast to prior work that often focuses solely on foreground object manipulation (MAHFOUDI et al., 2019; Novozamsky et al., 2020). An overview of our image manipulation pipeline is given in Figure 4 (Appendix).

Our MAGIC-dataset exhibits significant diversity in manipulation sizes. The MAGIC-News subset shows a wide range of manipulation sizes, from small edits to large alterations, requiring detection models to be robust across various levels of visual modifications. In contrast, manipulation sizes in MAGIC-COCO are skewed toward smaller edits, with relatively few large manipulations. Specifically, in MAGIC-News, there are 39,188 large manipulations covering > 70% of the image area,

---

[1]One exception is the International topic, which has fewer images available.

Table 2: Number of images per topic in our MAGIC-News subset.

| Business | Crime | Science | International | Arts | Entertainment | Politics | Media |
|---|---|---|---|---|---|---|---|
| 14,463 | 12,210 | 12,610 | 3,151 | 11,790 | 12,294 | 11,522 | 12,441 |

21,587 medium manipulations covering between 30% and 70% of the image area, and 15,124 small manipulations covering < 30% of the image area. For MAGIC-COCO, the breakdown is 76,157 small, 9,275 medium, and 1,602 large manipulations, respectively.

### 3.3 DATASET QUALITY SURVEY

To assess the quality of the manipulated images and ensure their practical use for manipulation detection, we perform a human evaluation via Amazon Mechanical Turk. A total of 4,950 images were used in our survey (2,750 from MAGIC-News and 2,200 from MAGIC-COCO). We include 5 generators for MAGIC-News and 4 for MAGIC-COCO[2]. We sample 500 images from each generator. Additionally we include 250 authentic images from MAGIC-News and 200 from MAGIC-COCO. Each image is evaluated by 3 people; in total we have 14,850 responses from 1,829 unique workers[3]. The respondents were shown two images: a manipulated image $X$ modified in a region specified by a binary mask $M$, and a copy of $X$ with a mask superimposed on it with a transparency value $\alpha$, as shown in Figure 6 in the Appendix. The respondents are asked the following questions: Q1) *"Do you think this image is manipulated?"* (yes or no), Q2) *"Do you see the **object** in the image (you can use the mask overlay to the right of the Image to better see the object)?"* (yes or no); Q3) *"Does the **object** look realistic?"* (yes, maybe or no), and Q4) *"Does the **object** look natural in the background?"* (yes, maybe, no). For each example, we specify which **object** the respondents should look for (e.g., a "giraffe"). We discard the answers to Q3 and Q4 if the respondent answered "No" to Q2, which results in 9,596 responses. Next, we utilize Q3 to create pseudo-labels for "High quality" and "Low quality" images. By combining the "Maybe" and "No" responses and utilizing majority voting, we get 1,507 "Low quality" images. Meanwhile, we have 1,905 "Yes" responses corresponding to "High quality" images, hence we get 3,412 images in total. We discuss the results of utilizing these pseudo-labels for comparison with selected manipulated detection models in Section 4.

## 4 EXPERIMENTS

We investigate two main aspects of image manipulation detection models using the proposed MAGIC dataset: (1) the generalization capabilities across different axes (e.g., image source), (2) the performance in context of human evaluation of diffusion-based manipulations.

### 4.1 EXPERIMENTAL SETUP

Our dataset contains 7 distinct manipulations, as detailed in Section 3. To benchmark the recent manipulation detection models, we have employed two different data splits based on manipulation type and image subset (News or COCO): one for in-distribution (ID) and another for out-of-distribution (OOD) performance assessment. We compare a number of different image manipulation detection models in our experiments, namely: **Swin** Transformer Liu et al. (2021) used as the base encoder for an Upernet model (Xiao et al., 2018), **PSCC-Net** (Liu et al., 2022), **HiFi** (Guo et al., 2023), **EVP** (Liu et al., 2023), **DOLOS** (Tântaru et al., 2024) and additionally utilize domain generalization methods like **SWAD** (Cha et al., 2021) and **Model Soups** (Wortsman et al., 2022) combined with EVP. Figure 3 outlines the distribution of manipulation types across the MAGIC-News and MAGIC-COCO subsets, categorizing them into methods used for training and those reserved for testing as out-of-domain. For the MAGIC-News subset, the in-domain (MT-ID) training methods include Blended-Diffusion, GLIDE, and Latent Diffusion. The out-of-domain (MT-OOD) testing methods for this subset are Stable Diffusion, GLIGEN Splicing, and Adobe Firefly. For the MAGIC-COCO subset, the in-domain training methods are the same as for MAGIC-News, while the out-of-domain testing

---

[2]This evaluation included all manipulation types with the exception of Adobe Firefly (for either subset) and Blended Latent Diffusion / GLIGEN (for MAGIC-COCO).

[3]The workers received $0.1 for completing the survey.

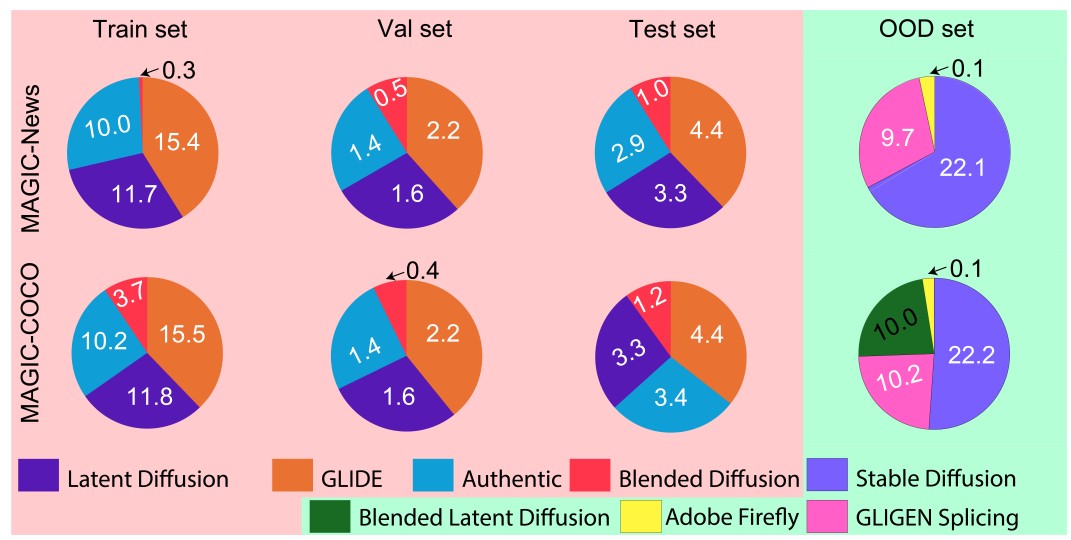

Figure 3: Breakdown of the manipulation types in MAGIC-News and MAGIC-COCO; numbers are in tens of thousands.

Table 3: Comparing the AUC and F1 score of models trained on MAGIC-News and MAGIC-COCO respectively and tested on MAGIC-News test set and OOD set or tested on MAGIC-COCO test set and OOD set to measure generalization across image sources and manipulation types.

| Trained on: | MAGIC-News | | | | MAGIC-COCO | | | | MAGIC-COCO | | | | MAGIC-News | | | |
|---|---|---|---|---|---|---|---|---|---|---|---|---|---|---|---|---|
| Tested on: | MAGIC-News | | | | | | | | MAGIC-COCO | | | | | | | |
| | MT-ID | | MT-OOD | | MT-ID | | MT-OOD | | MT-ID | | MT-OOD | | MT-ID | | MT-OOD | |
| | AUC | F1 | AUC | F1 | AUC | F1 | AUC | F1 | AUC | F1 | AUC | F1 | AUC | F1 | AUC | F1 |
| Swin | 65.9 | 63.8 | 57.0 | 44.5 | 54.3 | 21.6 | 50.1 | 0.9 | 60.6 | 31.6 | 57.3 | 15.5 | 50.1 | 0.7 | 49.9 | 0.0 |
| EVP | 79.2 | 73.4 | 61.9 | 49.0 | 62.0 | 49.2 | 55.7 | 44.9 | 79.0 | 50.7 | 79.6 | 38.4 | **62.1** | **23.9** | **67.6** | **25.2** |
| EVP+SWAD | 79.5 | 73.2 | 62.8 | **53.4** | 60.2 | 45.2 | 56.7 | 39.4 | 79.2 | 52.8 | **84.3** | 42.3 | 58.3 | 22.5 | 66.9 | 24.3 |
| EVP+Soup | **80.9** | 74.9 | **64.2** | 51.8 | 63.7 | 47.7 | 57.4 | 33.6 | **80.6** | **57.8** | 84.0 | **43.5** | 54.9 | 20.8 | 59.8 | 22.0 |
| DOLOS | 78.1 | 71.1 | 57.0 | 49.0 | **69.6** | **55.4** | **59.7** | **52.4** | 61.3 | 21.8 | 62.0 | 23.3 | 48.5 | 21.8 | 52.9 | 24.3 |
| PSCC-Net | 72.9 | 72.8 | 49.5 | 48.9 | 51.8 | 29.0 | 49.7 | 3.90 | 71.6 | 36.8 | 70.2 | 30.9 | 48.8 | 4.8 | 49.7 | 4.5 |
| HiFi | 73.6 | **77.8** | 50.9 | 29.9 | 49.6 | 10.1 | 48.6 | 1.9 | 66.8 | 34.6 | 62.7 | 21.5 | 51.5 | 5.7 | 51.6 | 7.6 |

methods additionally include Blended Latent Diffusion. The in-domain data for both MAGIC-News and MAGIC-COCO is further split into a 70%-10%-20% ratio for training, validation, and testing, respectively for each of the manipulation types.

## 4.2 GENERALIZATION PERFORMANCE

We propose three primary ways of evaluating model generalization across different types of domain shifts: image source generalization, manipulation type generalization, topic source generalization. Additionally, we explore the impact of the manipulation sizes. This approach allows us to comprehensively assess the robustness and adaptability of the models in diverse scenarios.

### 4.2.1 IMAGE SOURCE AND MANIPULATION TYPE GENERALIZATION

**Image Source Generalization.** First, we focus on generalization across different image sources. We train manipulation detection models on either MAGIC-News or MAGIC-COCO and test them both in-domain (ID) and on the other subset (OOD); this is done for both datasets. This setup allows us to measure how well the models generalize to data from a different source.

Table 4: Comparing the AUC performance of models trained on the MAGIC-News subset to generalize across 8 selected topics.

| Tested on: | Crime | Business | Science | Media | Entertainment | Arts | Politics | International |
|---|---|---|---|---|---|---|---|---|
| EVP | **75.5** | **71.4** | **65.8** | **68.9** | **75.6** | **69.9** | **74.7** | **72.0** |
| DOLOS | 68.1 | 65.9 | 59.3 | 59.0 | 67.0 | 60.3 | 60.4 | 65.1 |
| PSCC-Net | 53.5 | 52.1 | 57.1 | 58.1 | 53.2 | 58.5 | 60.6 | 55.8 |

**Manipulation Type Generalization.** The next angle we investigate is whether models generalize across different manipulation types. As mentioned in Figure 3, in-distribution manipulations (MT-ID) include methods such as Blended-Diffusion, GLIDE, and Latent Diffusion for both MAGIC-News and MAGIC-COCO. Out-of-distribution manipulations (MT-OOD), reserved for testing, include methods such as Stable Diffusion, GLIGEN Splicing, and Adobe Firefly for MAGIC-News, and Stable Diffusion, Blended-Latent Diffusion, GLIGEN Splicing and Adobe Firefly for MAGIC-COCO. Notably, the Adobe Firefly subset is distinct, as it includes data inpainted by the authors within the Adobe tool. Here we gain insights into the models' ability to handle unseen manipulations.

**Results.** Table 4.2.1 reports the results from training the models on MAGIC-News and MAGIC-COCO to make the comparisons fair across the models. Our evaluation focuses on both in-distribution (ID) and out-of-distribution (OOD) performance to assess the models' generalization capabilities.

For the image source generalization scenario, EVP trained on MAGIC-News outperforms the other models in the OOD case (columns 8 and 9), which could be attributed to its transformer-based architecture that is known to enhance generalization capabilities under distribution shifts Zhang et al. (2022). However, DOLOS shows superior OOD performance when trained on MAGIC-COCO for the OOD case (columns 4 and 5), which can be attributed to it's Xception(Chollet, 2017) backbone trained on ImageNet(Deng et al., 2009) images, which can be similar to the images of MAGIC-News. We note that the base EVP outperforms the EVP+SWAD and EVP+Soup which highlights the difficulty of the image source generalization context and emphasizes that utilizing a popular domain generalization method may not easily solve this task.

Lastly, for manipulation type generalization, we see that in columns 2-3, PSCC-Net and HiFi both trained on MAGIC-News for ID both models having a similar CNN based architecture, does not perform as well as EVP or DOLOS. This could be attributed to the choice of utilizing a smaller version of the HRNet backbone which they both utilize for it's feature extractor that works well for traditional manipulation detection (Liu et al., 2022; Guo et al., 2023) but may could possibly not perform well for diffusion based inpaintings. But with EVP which utilizes a transformer based backbone Segformer Xie et al. (2021) and DOLOS which has been shown to perform well for diffusion based inpaintings Țânțaru et al. (2024), it is not surprising that they perform the best. Surprisingly DOLOS trained on MAGIC-COCO does not perform well in columns 6-7 which can potentially be explained by the significant amount of small manipulations in MAGIC-COCO, which highlights a challenging aspect of the dataset, which requires models to be able to detect a number of small manipulations.

One important aspect of our dataset is that we created the two subsets (MAGIC-COCO and MAGIC-News) to have similar amounts of training examples. This did highlight an interesting aspect: the models trained on MAGIC-News seem to perform better on OOD samples from MAGIC-COCO potentially because of the more balanced amount of small,medium and larger images, with the more challenging dataset MAGIC-COCO containing a large amount of small manipulation .

### 4.2.2 TOPIC SOURCE GENERALIZATION

**Topic Sources.** Beyond data source and manipulation type generalization, we explore the generalization of models across different topics within the MAGIC-News dataset. We investigate how each of the models perform on each of the topics present in MAGIC-News and determine where models perform poorly on. This evaluation helps us understand how well the models generalize to different subject matters, which is crucial for applications involving diverse content.

**Results.** Table 4 the results across the different topics of the MAGIC-News dataset. For all the models we train and validate on 70% of the data and test on the remaining 30% of the data. We see that

Table 5: Comparing the AUC/Precision/Recall performance of models trained on the MAGIC-News and MAGIC-COCO respectively, to study generalization across image distributions and manipulation sizes. The top 3 rows are the results on MAGIC-News's ID test set and OOD set by getting the average of all the results in the specific size category., the last 3 rows of results are tested on MAGIC-COCO's ID test set and OOD set by getting the average of all the results in the specific size category *Small: $\leq 30\%$ coverage, $> 30\%$ *Medium $\leq 70\%$ *Large: $> 70\%$.

| Trained On: | MAGIC-News | | | MAGIC-COCO | | |
|---|---|---|---|---|---|---|
| Tested On: | MAGIC-News | | | | | |
| | AUC/Pre/Rec | AUC/Pre/Rec | AUC/Pre/Rec | AUC/Pre/Rec | AUC/Pre/Rec | AUC/Pre/Rec |
| | Small | Medium | Large | Small | Medium | Large |
| EVP | **78.9/44.8**/55.0 | **80.2/75.5**/65.1 | 58.7/93.9/39.4 | 64.7/24.2/0.09 | 58.1/38.2/0.07 | 49.8/39.4/0.02 |
| DOLOS | 72.5/38.5/66.7 | 76.4/71.0/76.6 | **66.3/95.1**/81.4 | **69.2/33.1/58.7** | **70.3/65.3/63.2** | **59.3/93.8/67.1** |
| PSCC-Net | 63.4/33.4/**71.5** | 69.4/62.5/**89.6** | 63.6/92.9/**98.0** | 51.1/0.06/0.07 | 51.0/14.3/0.09 | 50.5/19.4/0.05 |
| Trained on: | MAGIC-News | | | MAGIC-COCO | | |
| Tested On: | MAGIC-COCO | | | | | |
| | Small | Medium | Large | Small | Medium | Large |
| EVP | **67.9/16.8**/0.05 | **62.3**/38.6/0.03 | **57.3**/62.2/0.03 | **88.7/50.6**/44.0 | **87.3/87.1**/42.0 | 67.6/**89.2**/0.08 |
| DOLOS | 50.3/10.0/**19.9** | 49.5/**43.1/20.0** | 53.8/**86.5**/23.8 | 63.6/18.2/13.7 | 62.1/58.9/13.4 | 61.8/89.1/12.5 |
| PSCC-Net | 49.6/0.32/0.03 | 48.4/0.80/0.03 | 39.9/19.8/0.08 | 75.4/21.1/**57.7** | 71.6/58.0/**63.8** | **69.2**/88.1/**66.9** |

for EVP, it performs the best across all topics when compared to DOLOS and PSCC-Net, however when it compares to the individual topics themselves the performance has some stark differences. Take for instance the different between Entertainment and Science, we can see that there is almost a 10% difference between the AUC performance. When looking at the images one major difference would be the type of content in each of the images, with Entertainment images focuses more on people where a specific object is being focused in an image. This can make it clearer to determine which object is being manipulated as EVP utilizes the SegFormer transformer backbone which has been shown to perform well when detecting different objects for images in the wild (Xie et al., 2021). However for the Science images this can vary quite a bit, with varying objects being the focus of the image, for instance an image of a group of people might be shown but the manipulated object is a small car in the background. We observe that images related to Science can vary by quite a lot and may contain a number of objects which can confuse models as to what is manipulated. This highlights how images by topic can result in challenging manipulations that these manipulation detection models can struggle with at times because of the varying objects present in the images.

### 4.2.3 MANIPULATION SIZE GENERALIZATION

**Effect of Manipulation Sizes.** The final aspect of our dataset is the manipulation size. Here, we determine how SoTA models perform across different sizes of manipulations. We categorize small manipulated objects as those that take up $\leq 30\%$ of the size of the image, and medium objects as those that take up $> 30\%$ and $\leq 70\%$, large objects as those that take up $> 70\%$ of the image.

**Results.** Table 5 highlights the performance of our top performing models on small, medium and large manipulations. We can see that for EVP, when trained on MAGIC-News and MAGIC-COCO, the AUCs for small and medium manipulations are higher than for the large ones. When looking at the precision and recall for the large manipulations for EVP we can see that the precision is high but the recall is quite low. This trend carries over for OOD for both MAGIC-News and MAGIC-COCO, it even occurs for MAGIC-COCO for ID. Because of the higher performance for smaller manipulations for EVP this could be attributed to the transformer based backbone Segformer (Xie et al., 2021) which utilizes an MLP based decoder which could help the model perform better for the smaller manipulations. Due to the fact that the image size distribution is rather different for MAGIC-News versus MAGIC-COCO, this highlights another difference between the two, with MAGIC-COCO proving to be more difficult for even the better performing models like EVP.

Table 6: The data quality survey outcomes, split by the inpainting type. We report majority vote for each question. Q1) *"Do you think this image is manipulated?"*, Q2) *"Do you see the **object** in the image (you can use the mask overlay to the right of the Image to better see the object)?"*; Q3) *"Does the **object** look realistic?"*, and Q4) *"Does the **object** look natural in the background?"*

|  | MAGIC-News | | | | | MAGIC-COCO | | | |
|---|---|---|---|---|---|---|---|---|---|
|  | GLIGEN | Blended | GLIDE | Latent | Stable | Blended | GLIDE | Latent | Stable |
| Q1↓ | 84.0 | 84.0 | 80.4 | 81.6 | 79.2 | 81.2 | 80.6 | 84.2 | 79.4 |
| Q2↑ | 72.6 | 81.4 | 75.4 | 72.0 | 82.0 | 81.2 | 80.6 | 84.2 | 79.4 |
| Q3↑ | 48.5 | 51.0 | 59.5 | 54.6 | 54.1 | 57.1 | 60.8 | 54.8 | 61.2 |
| Q4↑ | 56.1 | 54.2 | 63.1 | 59.5 | 65.2 | 57.7 | 61.4 | 55.8 | 62.2 |

Table 7: The analysis of the model performance (AUC) based on our data quality survey outcomes. We train our models on respective subsets and test them in domain on 1,121 images for MAGIC-Mews and 1,152 for MAGIC-COCO (see main text for discussion).

| Tested on: | MAGIC-News | | | MAGIC-COCO | | |
|---|---|---|---|---|---|---|
| Model | Blended | GLIDE | Latent | Blended | GLIDE | Latent |
| EVP | 90.5 | 88.2 | 88.6 | 79.2 | 90.9 | **86.1** |
| DOLOS | 95.9 | 89.5 | **93.4** | 69.3 | 70.3 | 61.6 |
| PSCC-Net | **97.3** | **94.8** | 85.0 | **95.7** | **92.7** | 68.5 |

## 4.3 HUMAN EVALUATION ANALYSIS

When looking at Table 6 we can see that for Q3, GLIGEN Splicing, Blended and Stable diffusion for News tended to be the worst performing model with a number of people realizing that those images were manipulated. On average for Q3 it appears that MAGIC-COCO tended to perform better with their manipulations for instance with GLIDE and Stable Diffusion being the top performing models. Q1 reveals that most persons did realize the image were manipulated which is to be expected as diffusion based inpaintings are still not perfect on average.

In Section 3.3 we explained how we selected 3,412 images. Next, we determine which images belong to the MAGIC-News and MAGIC-COCO ID test sets, and obtain 1,121 images for MAGIC-News and 1,152 for MAGIC-COCO. Table 7 highlights the performance of the models trained in-domain and tested on these images. When comparing these results with those from Table 6 we see that even thought EVP and DOLOS tended to be the better performing models, when tested on MAGIC-COCO. They performed significantly worse than PSCC-Net for Blended Diffusion which could be explained by the smaller amount of Blended Diffusion examples in our training data as shown in Figure 3 combined with the difficults of MAGIC-COCO in general. On average we see that models performed worse on MAGIC-COCO in Table 7 versus MAGIC-News and this follows with what we see in Table 6 for Q3, as persons agreed that the manipulations from MAGIC-COCO were better on average. Hence we have shown that MAGIC-COCO is a more challenging dataset than MAGIC-News for diffusion based inpainting as the human based evaluation corresponds with our findings from Section 4.2.1and therefore is a promising dataset for manipulation detection models to be tested on.

## 5 CONCLUSION

We introduced Multi-domain Analysis and Generalization of Image manipulation loCalization (MAGIC), a large image manipulation dataset aimed at studying the robustness and generalization capabilities of image manipulation detectors. Our dataset features a range of image sources, topics, manipulation types, and sizes. Notably, we have utilized state-of-the-art image manipulation techniques. Through extensive experiments, we found that while current detectors perform well on in-distribution data, they struggle on out-of-distribution samples, underscoring the need for better generalization. In our future research we will focus on integrating further contextual information such as image captions and news articles into our study.

## 6 ETHICS STATEMENT

Our work focuses on benchmarking and advancing the methods for detecting manipulated images. We believe this is an important effort aimed at countering misinformation online, especially in the era of advanced generative models. Thus, this work is ethical by its nature. At the same time, since we are producing a dataset with manipulated images, there is some potential for misuse, i.e., it being used for training even more sophisticated falsification methods. Besides, since we are relying on generative models to create the data, there is a possibility of biases inherent to these models propagating into our generated manipulations. Any potential users of our dataset should be mindful of that.

## 7 REPRODUCIBILITY STATEMENT

We intend to release our dataset and make our experiments reproducible by providing all the necessary information on a project page (yet to be established) upon paper acceptance. The copyright and usage rights of the VisualNews images are subject to that of Liu et al. (2020).

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

Figure 4: For MAGIC-News, we randomly select a region or mask generated by Mask2Former (Cheng et al., 2022). In the case of MAGIC-COCO, we use the provided ground-truth masks and select one at random. The selected region, along with its corresponding class label, is then passed to a manipulation model, which generates an altered version of the sample. See Section 3 for details.

Dragoș-Constantin Tântaru, Elisabeta Oneață, and Dan Oneață. Weakly-supervised deepfake localization in diffusion-generated images. In *Proceedings of the IEEE/CVF Winter Conference on Applications of Computer Vision*, pp. 6258–6268, 2024.

# A  APPENDIX

In this supplementary we include additional information about our experiments: 1) We conduct another analysis of our human eval results by looking at the High versus Low quality images. 2) We utilize the captions that are included with our MAGIC-News and MAGIC-COCO dataset to determine if their is any discrepancy between captions that are related or not to the manipulated object. 3) We include an example image of what a human annotator would have seen during the human evaluation. 4) An example of of a high level view of our image manipulation pipeline. Additionally, we include an example of what the manually create Adobe Firefly manipulations looked like before and after.

## A.1  IMAGE MANIPULATION PIPELINE

A high level view of our image manipulation manipulation pipeline can be seen in Figure 4

## A.2  EDITING TYPE BREAKDOWN

A high level view of our editing types statistics can be seen in Table 10

## A.3  HUMAN EVALUATION QUALITY EXPERIMENT

**High quality versus Low quality experiment** Following the labelling of the "High" and "Low" quality images from our human evaluation, we have tested our models on this data as seen in Table 8, this highlights that the ranking of models based on the human evaluation results did not change the order of the models by much. On average we can see that for EVP generally remains the top performing model as seen in in columns 2-3,6-9 except of the case of being trained on MAGIC-COCO and being tested on MAGIC-News in columns 4-5. However, this phenomena is also showcased in Table 4.2.1 which highlights an important aspect of our dataset where training on one image source does not guarantee high performance on another image source. This shows that our dataset is inline with how human evaluators would look at the manipulated images and highlights the importance of training and testing these manipulation segmentation models to better combat against diffusion base inpaintings.

Table 8: Comparing the AUC results from our human evaluation by utilizing majority voting we use 3412 images from our test and OOD set based on question 3 and combined answer choice Maybe with No. An image is considered of "High" quality if more people consider it realistic and "Low" if more people consider it not-realistic. The first 4 columns of results are tested on MAGIC-News's ID test set and OOD set combined and the last 4 columns of results are tested on MAGIC-COCO's ID test set and OOD set combined.

| Trained on: | MAGIC-News | | MAGIC-COCO | | MAGIC-COCO | | MAGIC-News | |
|---|---|---|---|---|---|---|---|---|
| Model | Low | High | Low | High | Low | High | Low | High |
| EVP | 80.9 | 80.7 | 58.9 | 59.5 | 90.3 | 88.0 | 69.3 | 68.0 |
| DOLOS | 80.3 | 79.8 | 75.4 | 75.1 | 66.1 | 64.5 | 66.1 | 64.5 |
| PSCC-Net | 72.8 | 71.8 | 47.0 | 48.9 | 78.2 | 77.2 | 49.3 | 49.0 |
| HiFi | 72.6 | 72.3 | 48.4 | 48.7 | 73.3 | 71.5 | 51.5 | 51.4 |

Table 9: Comparing the AUC performance of models trained on the MAGIC-News and MAGIC-News respectively subset to generalize across image distributions and caption relevance. *Cap-Ref: manipulated object in caption *Not Ref: manipulated object not in caption. Columns 2-5 are tested on the MAGIC-News test set and columns 6-9 are tested on the MAGIC-COCO test set.

| | MAGIC-News | | MAGIC-COCO | | MAGIC-News | | MAGIC-COCO | |
|---|---|---|---|---|---|---|---|---|
| | Cap-Ref | Not Ref | Cap-Ref | Not Ref | Cap-Ref | Not Ref | Cap-Ref | Not Ref |
| EVP | **75.3** | **75.0** | 60.7 | 57.1 | **89.6** | **86.1** | **66.2** | **67.5** |
| DOLOS | 72.6 | 71.8 | **67.9** | **66.5** | 64.1 | 63.1 | 49.8 | 50.5 |
| PSCC-Net | 65.2 | 64.9 | 51.2 | 50.2 | 75.6 | 74.5 | 49.2 | 49.4 |

## A.4 SEMANTIC SALIENCY

**Manipulation Semantic Salience** Another aspect of our dataset is looking at the semantic saliency with respect to captions, namely if a manipulated object is mentioned in a caption describing a manipulated image. For this task we utilize the original captions used from Visual News and COCO.

**Generalizing across manipulation semantic salience** Table 9 refers to the results obtained for related and unrelated captions with respect to the manipulated object being mentioned in the caption. We can see a similar trend to Table 4.2.1 whereby for both related and unrelated captions EVP tends to be the best performing model for columns 2-3 and 6-9 as to be expected with DOLOS performing the best for columns 4-5. For EVP, columns 4-5 and 6-7 we can see that the *Cap-Ref were the only times it scored slightly higher than *Not Ref, we know for MAGIC-News there are more related captions hence the higher performance for columns 4-5 are expected. However for columns 6-7, being tested on the MAGIC-COCO test set showcases again how challenging the MAGIC-COCO subset can be, as we can see there are less related captions as highlighted in Section 3.2. Hence, having a model that has a loosely related caption can possibly highlight challenging manipulations to detect even with the best performing manipulation detection models.

## A.5 HUMAN EVALUATION DETAILS

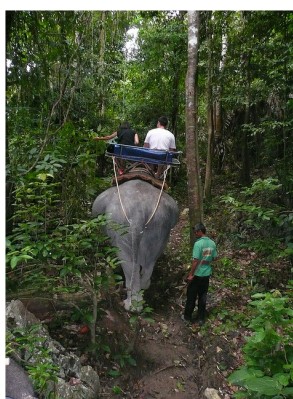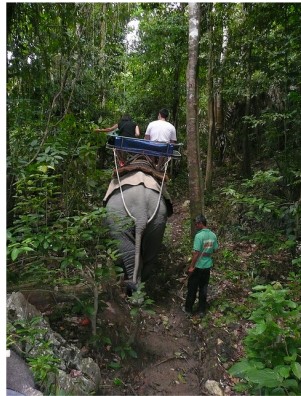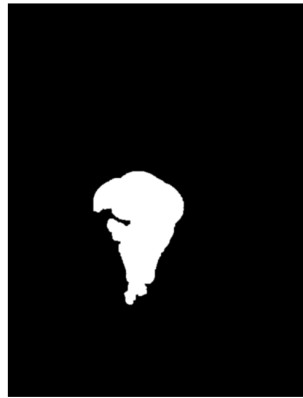

Figure 5: An example of a Adobe Firefly inpainted image from COCO described in section 3.1. With the left most being the original and the middle being the Firefly Inpainted image.

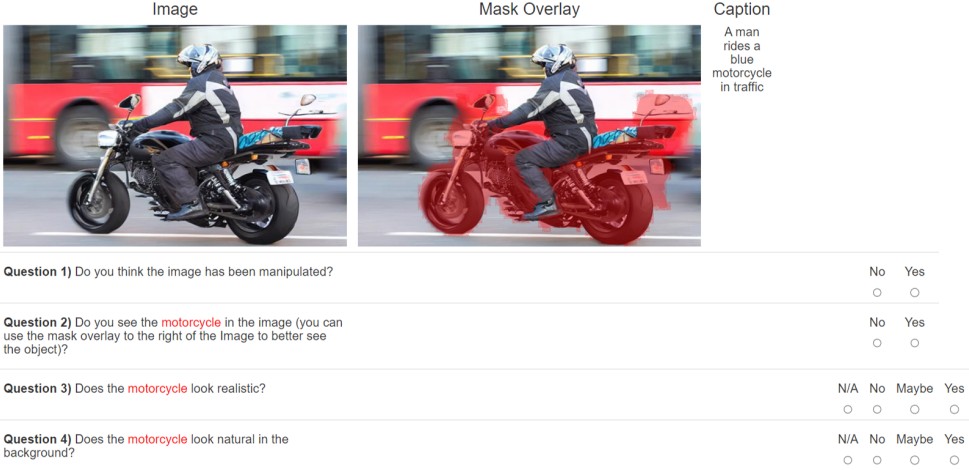

Figure 6: An example of an image from our dataset which human evaluators were given to answer questions on.

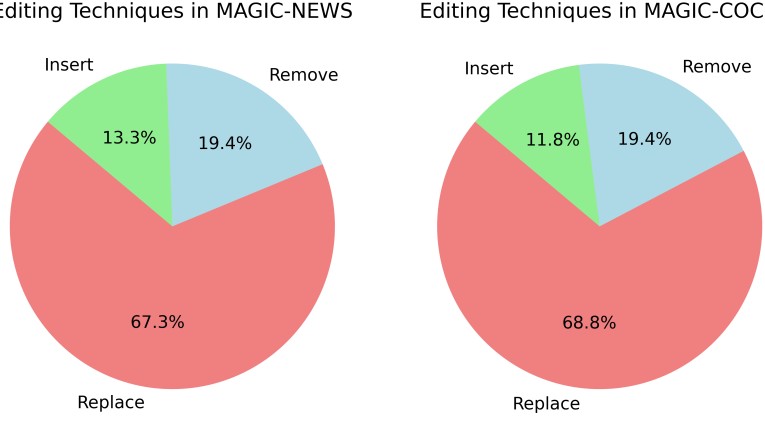

Figure 7: Visualization of manipulation sizes across different editing techniques from in fig. 2.

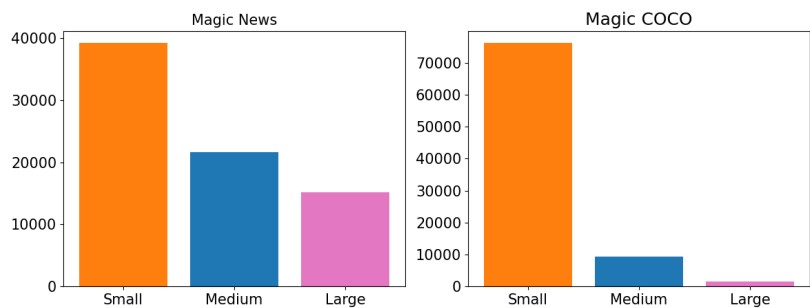

Figure 8: Visualization of manipulation sizes across different image sources from fig. 2.

Table 10: Distribution of Editing Techniques across MAGIC-News and MAGIC-COCO. Please refer to Figure 3 for a more detailed breakdown of the number of images for MAGIC-COCO and MAGIC-News based on manipulation type. *SD: Stable Diffusion (Rombach et al., 2022b), *BD: Blended Diffusion (Avrahami et al., 2022), *GLIDE (Nichol et al., 2022), *BLD: Blended Latent Diffusion (Avrahami et al., 2023), *AF: Adobe Firefly (Adobe, 2024), *LD: Latent Diffusion (Rombach et al., 2021), *GLIGEN (Li et al., 2023). *Blended-Latent Diffusion only occurs in the Out-Of-Distribution set of MAGIC-COCO

| Editing Technique | Number of Images | | Manipulation Type |
|---|---|---|---|
| | MAGIC-News | MAGIC-COCO | |
| Replacement | 49,493 | 59,824 | SD, BD, GLIDE, BLD, AF |
| Removal | 9,746 | 10,290 | GLIGEN |
| Insertion | 14,205 | 16,854 | LD |