# OpenReview forum: "Multi-domain Analysis and Generalization of Image manipulation loCalization"
_ICLR.cc/2025/Conference — Submitted to ICLR 2025_

### Official Review · Reviewer_VNC9 · 2024-10-31

**Soundness:** 2
**Presentation:** 2
**Contribution:** 2
**Rating:** 6
**Confidence:** 4

**Summary:**

This paper introduces two novel datasets for image forensics specifically curated from diffusion-based editing methods: MAGIC-News and MAGIC-COCO. These datasets encompass various topics and object classes, with manipulations including object insertion, replacement, and removal, applied through various editing techniques such as Stable Diffusion, Blended Diffusion, Glide Diffusion, and Adobe Firefly. Experiments demonstrate the performance of several image forensic techniques on these new datasets.

**Strengths:**

+ Propose new datasets that cover many situations: news (MAGIC-News) or daily lives (MAGIC-COCO)
+ Cover many editing operations: insertion, removal, and replacement
+ Include many editing techniques: Stable Diffusion, Blended Diffusion, Glide Diffusion, and Adobe Firefly.

**Weaknesses:**

Since this paper mainly focuses on new datasets, the presentation of the datasets should be prepared more carefully. Specifically:
+ Lack of high-quality examples of manipulated images and corresponding GT segmentation mask (Fig. 2 presents low-resolution images)
+ Instead of just listing some numbers only, the visual charts should be used to summarize the statistics of datasets (e.g., editing areas statistics
+ A table of Editing technique summarization would also help, including the number of images, and examples.
+ For the Dataset Quality Survey, using a flowchart to visualize the process would be better.
+ Lack of reporting IoU (along with AUC)
+ Lack of classification performance (decide whether an image is a manipulated image or genuine one)
+ The quality of the proposed dataset is a concern (Tab. 6) since some methods are just around 50%)
+ Demonstration of using the proposed datasets would help the performance of detection techniques on other datasets such as MagicBrush [a] and CocoGLIDE [b].

[a] Kai Zhang, Lingbo Mo, Wenhu Chen, Huan Sun, and Yu Su. Magicbrush: A manually annotated dataset for instruction-guided image editing. Advances in Neural Information Processing Systems, 36, 2024. 2, 5, 6, 7278

[b] Fabrizio Guillaro, D Cozzolino, Avneesh Sud, Nick Dufour, and L Verdoliva. TruFor: Leveraging all-round clues for trustworthy image forgery detection and localization. Proc. IEEE Comput. Soc. Conf. Comput. Vis. Pattern Recognit., pages 20606–20615, December 2022

**Questions:**

+ How to make sure the quality of the generated masks for MAGIC-News (since they are generated automatically from Mask2Former)
+ For the replacement operation, have you tested different-class replacements instead of same-class replacements
+ How to ensure the quality of GLIGEN since it is not a perfect method, any mechanism to ensure its quality?
+ How many images are used for training, val, test, and out-of-domain subsets?

---

> ### Author Response · Authors · 2024-11-23
>
> We thank the reviewer for spending the time on this paper and giving us feedback, please see our replies below.
>
> >Lack of high-quality examples of manipulated images and corresponding GT segmentation mask (Fig. 2 presents low-resolution images)
>
> >The quality of the proposed dataset is a concern (Tab. 6) since some methods are just around 50%)
>
>
> Thank you for your feedback. Our dataset is indeed mixed in terms of the quality of the manipulations images due to limitations of these manipulation methods.  However, our goal is not to produce high-quality manipulations, but rather to understand how detection methods perform across various axes of generalization including manipulation type and different image domains.  Thus, a low quality manipulation that goes unnoticed by a detector is also a concern. For example, consider the impact low-quality manipulations would have on moderation efforts in a form.  Since they are easy to identify, they would likely get flagged by these users rather than flagged by a manipulation detector, which would identify it as authentic.  Due to this conflict between the users and automatic detectors, a moderator may be tasked with reviewing the image as well, expending costly resources.
>
> What’s more, in Table 8 our work also highlights that manipulation detectors find low and high quality manipulations equally challenging, with little difference between the two sets. This illustrates an important observation made by our work: human judgements and machine detectors do not use similar evidence to identify manipulations.  This is to be expected to some degree, as automatic metrics judging image manipulation and generation quality are challenging to construct, making human judgements the gold standard for those tasks.  Thus, we show the quality of the manipulations has little to do with the goal of our work: creating high quality manipulation detection methods that generalize across a range of distribution shifts.
>
>
> >Instead of just listing some numbers only, the visual charts should be used to summarize the statistics of datasets (e.g., editing areas statistics
>
> We are preparing these visualizations and will update you once they are included in the paper.
>
> >A table of Editing technique summarization would also help, including the number of images, and examples.
>
> Below is a table with the summarized numbers of images per technique, we refer to Figure 3 for a more detailed breakdown of the number of images for Magic-COCO and Magic-News based on manipulation type. We added it to the Appendix in Table 10. For illustrative examples for the same, please see Figure 2.
>
> |                   | Magic-News       | Magic-COCO       |                                                                                               |
> |-------------------|------------------|------------------|-----------------------------------------------------------------------------------------------|
> | Editing Technique | Number of Images | Number of Images | Inpainting Techniques                                                                         |
> | Replacement       | 49493            | 59824            | Stable-Diffusion,Blended-Diffusion,Glide-Diffusion,*(Blended-Latent Diffusion), Adobe Firefly |
> | Insertion         | 9746             | 10290            | GLIGEN Splicing                                                                               |
> | Removal           | 14205            | 16854            | Latent Diffusion                                                                              |
> *Blended-Latent Diffusion only occurs in the Out-Of-Distribution set of Magic-COCO

---

> ### Author Response · Authors · 2024-11-23
>
> >For the Dataset Quality Survey, using a flowchart to visualize the process would be better.
>
> The survey itself is simply a questionnaire and not a process, an example of which is provided in Figure 6 of our paper.
>
> >Lack of reporting IoU (along with AUC)
>
> We have also added the F1 scores to Table 3, partly reproduced below with IoU scores as well, as you suggested. We observed that the F1 and IoU scores are also generally low across both manipulation OOD and image source OOD settings. The observed scores, particularly the low AUC (along with F1 and IoU) values in certain cases, reflect the core focus of our paper: highlighting that current models struggle to generalize across different domains for the problem of image manipulation detection.
>
> | Trained on | Magic-News     |                | Magic-COCO     |                | Magic-COCO     |                | Magic-News     |                |
> |------------|----------------|----------------|----------------|----------------|----------------|----------------|----------------|----------------|
> | Tested on  |                | Magic-News     |                |                |                | Magic-COCO     |                |                |
> |            | MT-ID          | MT-OOD         | MT-ID          | MT-OOD         | MT-ID          | MT-OOD         | MT-ID          | MT-OOD         |
> |            | AUC/F1/IoU     | AUC/F1/IoU     | AUC/F1/IoU     | AUC/F1/IoU     | AUC/F1/IoU     | AUC/F1/IoU     | AUC/F1/IoU     | AUC/F1/IoU     |
> | EVP        | 79.2/73.4/61.6 | 61.9/49.0/26.7 | 62.0/49.2/14.7 | 55.7/44.9/3.39 | 79.0/50.7/35.8 | 79.6/38.4/20.6 | 62.1/23.9/2.66 | 67.6/25.2/3.85 |
> | EVP + SWAD | 79.5/73.2/55.7 | 62.8/53.4/22.9 | 60.2/45.2/11.7 | 56.7/39.4/1.77 | 79.2/52.8/37.2 | 84.3/42.3/22.6 | 58.3/22.5/1.53 | 66.9/24.3/3.27 |
> | EVP + Soup | 80.9/74.9/63.8 | 64.2/51.8/29.2 | 63.7/47.7/17.4 | 57.4/33.6/3.63 | 80.6/57.8/43.4 | 84.0/43.5/23.1 | 54.9/20.8/3.77 | 59.8/22.0/1.25 |
> | DOLOS      | 78.1/71.1/61.4 | 57.0/49.0/34.4 | 69.6/55.4/36.6 | 59.7/52.4/38.2 | 61.3/21.8/2.59 | 62.0/23.3/6.63 | 48.5/21.8/3.39 | 52.9/24.3/9.39 |
> | PSCC-Net   | 72.9/72.8/61.6 | 49.5/48.9/37.5 | 51.8/29.0/8.91 | 49.7/3.90/0.44 | 71.6/36.8/23.6 | 70.2/30.9/17.5 | 48.8/4.78/0.30 | 49.7/4.50/0.32 |
> | HiFi       | 73.6/77.8/70.9 | 50.9/29.9/21.4 | 49.6/10.1/6.45 | 48.6/1.85/1.09 | 66.8/34.6/24.0 | 62.7/21.5/14.2 | 51.5/5.73/3.71 | 51.6/7.55/4.51 |
>
>
> >Lack of classification performance (decide whether an image is a manipulated image or genuine one)
>
> Following [k,l,m], we focus on the task of manipulation localization, whereas manipulation classification is an entirely different task.  That said, since some (but not all) models are designed to accomplish both tasks, we will update our paper with these results as soon as we have them.  Once that occurs, we will update you with those results.
>
> [k] Liu, Weihuang, et al. "Explicit visual prompting for low-level structure segmentations." Proceedings of the IEEE/CVF Conference on Computer Vision and Pattern Recognition. 2023.
>
> [l] Hao, Jing, et al. "Transforensics: image forgery localization with dense self-attention." Proceedings of the IEEE/CVF International Conference on Computer Vision. 2021.
>
> [m] Zhou, Jizhe, et al. "Pre-training-free image manipulation localization through non-mutually exclusive contrastive learning." Proceedings of the IEEE/CVF International Conference on Computer Vision. 2023.
>
> >Demonstration of using the proposed datasets would help the performance of detection techniques on other datasets such as MagicBrush [a] and CocoGLIDE [b].
>
> Based on this feedback we have begun testing a subset of our models on these datasets and once we have the results we will update this comment.

---

> > ### Author Response · Authors · 2024-11-26
> >
> > >Lack of classification performance (decide whether an image is a manipulated image or genuine one)
> >
> > As noted earlier, [k,l,m], we focus on the task of manipulation localization, whereas manipulation classification is an entirely different task.  However, to address this comment, as we believe it adds value, we evaluated this classification task by utilizing PSCC-Net and HiFi, since these both contain a classification head. We can see that PSCC-Net does better than HiFi in most cases even though their architecture is quite similar. Interestingly for PSCC-Net when trained on MAGIC-COCO and tested on MAGIC-COCO it performs quite well. One reason would be the HRNet backbone that PSCC-Net utilizes which was trained on ImageNet. However, both report a significant drop in performance when asked to generalize to OOD images, highlighting the importance of our work as we highlight these challenges.
> >
> > | Trained on | Magic-News | Magic-COCO | Magic-COCO | Magic-News |
> > |------------|------------|------------|------------|------------|
> > | Tested On  | Magic-News |            | Magic-COCO |            |
> > |            | AUC/F1     | AUC/F1     | AUC/F1     | AUC/F1     |
> > | PSCC-Net   | 69.0/64.1  | 40.7/37.4  | 93.0/90.1  | 66.4/63.7  |
> > | HiFi       | 54.7/54.7  | 50.7/34.9  | 56.0/55.8  | 53.9/53.6  |
> >
> > [k] Liu, Weihuang, et al. "Explicit visual prompting for low-level structure segmentations." Proceedings of the IEEE/CVF Conference on Computer Vision and Pattern Recognition. 2023.
> >
> > [l] Hao, Jing, et al. "Transforensics: image forgery localization with dense self-attention." Proceedings of the IEEE/CVF International Conference on Computer Vision. 2021.
> >
> > [m] Zhou, Jizhe, et al. "Pre-training-free image manipulation localization through non-mutually exclusive contrastive learning." Proceedings of the IEEE/CVF International Conference on Computer Vision. 2023.

---

> ### Author Response · Authors · 2024-11-23
>
> >How to make sure the quality of the generated masks for MAGIC-News (since they are generated automatically from Mask2Former)
>
> For our human evaluation because we have included Magic-News, this was a way for us to check how the manipulated objects looked overall, as we can see from question 1 and question 3, Magic-News which had generated masks performed around the same as Magic-COCO, which has ground truth segmentation masks. This suggests that there is little difference in the quality of the masks.
>
> >For the replacement operation, have you tested different-class replacements instead of same-class replacements
>
>
> We did not test different-class replacements in order to keep the semantic content of the image unchanged. However based on this feedback we will run a small scale experiment to determine the image it has on the models and once we have finished this experiment we will update this comment
>
> >How to ensure the quality of GLIGEN since it is not a perfect method, any mechanism to ensure its quality?
>
> GLIGEN’s ability to use spatially-aligned condition maps provides a unique framework to insert objects into specific locations within an image. Based on our experiments, the model often captures enough semantic and spatial information to create meaningful test cases for evaluating manipulation localization tasks. Specifically, in the human evaluation results we see that GLIGEN does perform similarly to other inpainting techniques like Blended Diffusion for question 3 (Does the object look realistic?) as reported in Table 6 with GLIGEN scoring 48.5 and Blended Diffusion scoring 51.0
>
>
> >How many images are used for training, val, test, and out-of-domain subsets?
>
> We have created a further breakdown of the results for each of the manipulations, this is similar to the information shown in Figure 3.
> |                   | MAGIC-News |      |       |                               |       |
> |:-----------------:|:----------:|:----:|:-----:|:-----------------------------:|:-----:|
> |                   | Train      | Val  | Test  |                               |       |
> |                   |    IM-ID   |      |       |             IM-OOD            |       |
> | Latent-Diffusion  | 11732      | 1676 | 3352  | Stable Diffusion              | 22125 |
> | GLIDE             | 15410      | 2201 | 4403  | Auto-Splice (GLIGEN Splicing) | 9746  |
> | Blended Diffusion | 3677       | 526  | 1051  |                               |       |
> | Authentic         | 10208      | 1458 | 2916  |                               |       |
> |                   |            |      |       | Adobe Firefly                 | 100   |
> | Total: 58610      | 41027      | 5861 | 11722 | Total: 31971                  | 31971 |
>
>
>
>
>
>
>
>
>
> |                   | MAGIC-COCO |      |       |                               |       |
> |:-----------------:|:----------:|:----:|:-----:|:-----------------------------:|:-----:|
> |                   | Train      | Val  | Test  |                               |       |
> |                   |    IM-ID   |      |       |             IM-OOD            |       |
> | Latent-Diffusion  | 11806      | 1676 | 3370  | Stable Diffusion              | 22251 |
> | GLIDE             | 15519      | 2201 | 4462  | Auto-Splice (GLIGEN Splicing) | 10290 |
> | Blended Diffusion | 3759       | 400  | 1212  | Blended-Latent Diffusion      | 10088 |
> | Authentic         | 10208      | 1458 | 3416  |                               |       |
> |                   |            |      |       | Adobe Firefly                 | 100   |
> | Total: 59487      | 41292      | 5735 | 12460 | Total: 42729                  | 42729 |

---

> ### Author Response · Authors · 2024-11-26
>
> >Instead of just listing some numbers only, the visual charts should be used to summarize the statistics of datasets (e.g., editing areas statistics
>
> Based on this feedback we have added two other visual charts to make it easier to see more statistics of our dataset based on editing areas and manipulations sizes adapted from the statistics we described in Section 3.2 of our paper.
>
> We have drafted two diagrams to address this comment. The first diagram is located in the Appendix as Figure 7 and it shows manipulation area sizes vs. editing techniques: This chart illustrates the distribution of manipulation area sizes across the three editing techniques: removal, insertion, and replacement.
> The second diagram is located in the Appendix as Figure 8, it is manipulation area sizes vs. image source: This chart compares the sizes of manipulated areas across the two sources, MAGIC-News and MAGIC-COCO.
>
> >Demonstration of using the proposed datasets would help the performance of detection techniques on other datasets such as MagicBrush [a] and CocoGLIDE [b].
>
> We conducted an experiment whereby we trained on MAGIC-News and tested on MagicBrush, reported below. As seen in the table below EVP performs quite well when trained on MAGIC-News and evaluated on Magic-Brush.
>
> | MagicBrush | AUC/F1/IoU     |
> |------------|----------------|
> | PSCC-Net   | 44.3/26.6/12.3 |
> | DOLOS      | 58.8/28.0/15.8 |
> | EVP        | 76.1/37.6/23.4 |
>
>
>
> For CocoGLIDE, we use the same three models trained on MAGIC-News, making CocoGLIDE images OOD. This is different from, say, the PSCC-Net reported in CocoGLIDE as that model is trained on 380K manipulated and pristine images extracted from COCO, whereas we use less than 100K news images.  That said, as shown below, PSCC-Net trained on our dataset reports similar performance as the model reported in CocoGlide. What’s more, the EVP outperforms even the model proposed in the CocoGLIDE paper (TruFor).  This helps illustrate the benefits that stem from using our dataset in other settings.
>
> | CocoGLIDE | AUC/F1/IoU     |
> |------------|----------------|
> | PSCC-Net (reported in CocoGLIDE)  | 77.7/51.5/- |
> | TruFor (proposed in CocoGLIDE)  | 75.2/52.3/- |
> | PSCC-Net   | 78.0/51.3/38.2 |
> | DOLOS      | 55.6/38.5/25.3 |
> | EVP        |  83.6/57.0/42.9 |
>
> >For the replacement operation, have you tested different-class replacements instead of same-class replacements
>
> As we discussed earlier, we did not test different-class replacements in order to keep the semantic content of the image unchanged. However based on this feedback we ran a small scale experiment to determine the impact the image class has on the models. Using 100 images from Magic-News we created images that had new objects replace old objects, for instance we replaced a car with a bike and so on to try and keep the same semantic relevance with the initial object. The same was done with Magic-COCO, and we also included the experiment with replacing the object with the same object like we have done in all our experiments (e.g. replace bike with bike). We can see that overall for both EVP and DOLOS the difference between the replacing the object with a new one, or replacing the object with the same object type has similar scores.
>
> | Trained on |   MAGIC-News   |    MAGIC-News   |   MAGIC-COCO   |    MAGIC-COCO   |
> |:----------:|:--------------:|:---------------:|:--------------:|:---------------:|
> |  Tested on |   MAGIC-News   |    MAGIC-News   |   MAGIC-News   |    MAGIC-News   |
> |            |   New-Object   | Replaced-Object |   New-Object   | Replaced-Object |
> |            |   AUC/F1/IoU   |    AUC/F1/IoU   |   AUC/F1/IoU   |    AUC/F1/IoU   |
> |     EVP    | 59.1/56.2/1.60 |  58.0/55.8/1.85 | 60.6/51.1/10.0 |  61.9/50.6/10.8 |
> |    DOLOS   | 54.1/54.6/5.72 |  55.0/54.9/6.20 | 51.5/58.3/2.66 |  47.7/58.4/2.15 |
>
> | Trained on |   MAGIC-COCO   |    MAGIC-COCO   |   MAGIC-News   |    MAGIC-News   |
> |:----------:|:--------------:|:---------------:|:--------------:|:---------------:|
> |  Tested on |   MAGIC-COCO   |    MAGIC-COCO   |   MAGIC-COCO   |    MAGIC-COCO   |
> |            |   New-Object   | Replaced-Object |   New-Object   | Replaced-Object |
> |            |   AUC/F1/IoU   |    AUC/F1/IoU   |   AUC/F1/IoU   |    AUC/F1/IoU   |
> |     EVP    | 87.0/53.1/35.0 |  86.9/53.2/34.8 | 60.8/26.0/1.81 |  61.8/25.9/0.77 |
> |    DOLOS   | 67.2/25.0/2.93 |  66.8/26.1/2.85 | 44.8/23.7/1.62 |  46.1/23.9/1.21 |

---

> ### Comment · Reviewer_VNC9 · 2024-11-26
>
> I want to thank the authors for their detailed responses. Most of my concerns regarding the presentation of the paper have been addressed. I hope the authors will include the new experiments and analyses in the revised version. However, I share Reviewer GsP7’s concern about the quality of the manipulated images in the proposed datasets. Despite the authors’ claims about the datasets covering various dimensions, including editing types and diverse domains, the image quality appears to be quite low. Given that these datasets are based on manipulated images generated by diffusion-based editing methods, they should ideally exhibit a level of quality comparable to that of diffusion-generated images. I will increase my score from 5 to 6.

---

> > ### Author Response · Authors · 2024-12-01
> >
> > First we would like to thank you for putting the time to participate in the discussion period for our paper and we thank you for raising your score. We know that the period to revise our pdf has passed, but we will add our experiments and analysis for the camera ready once our paper is accepted.

---

### Official Review · Reviewer_GsP7 · 2024-11-02

**Soundness:** 3
**Presentation:** 3
**Contribution:** 2
**Rating:** 6
**Confidence:** 3

**Summary:**

This paper proposed a new image manipulation location benchmark for diffusion-based generation methods. It contains two image sources and seven manipulation techniques. The experiments under several settings, also provide some interesting insights.

**Strengths:**

1. The proposed datasets seem good for image manipulation location tasks.
2. The writing and experiments are pretty good.

**Weaknesses:**

1. The quality of the manipulated images in Figure 2 is worrying, especially for the removing class. Although the authors mentioned that they apply human evaluation for the generated images, I'm worried about the data balance of the three categories (removal, replacement, and insertion) under high-quality annotations.
2. The image manipulation methods used in the paper are not very new. Using SD series, like SD2, or even SDXL is better.
3. In Table 3, it's interesting that the OOD score is higher than the ID score when trained on MAGIC-News and tested on MAGIC-COCO. It's better to provide a more depth analysis.

**Questions:**

Please see the weaknesses.

---

> ### Author Response · Authors · 2024-11-23
>
> We thank the reviewer for spending the time on this paper and giving us feedback, please see our replies below.
>
> >1) The quality of the manipulated images in Figure 2 is worrying, especially for the removing class … Even with a human evaluation, I’m worried about the three categories (removal, replacement, and insertion) under high-quality annotations.
>
> Thank you for your feedback. Our dataset is indeed mixed in terms of the quality of the manipulations images due to limitations of these manipulation methods.  However, our goal is not to produce high-quality manipulations, but rather to understand how detection methods perform across various axes of generalization including manipulation type and different image domains.  Thus, a low quality manipulation that goes unnoticed by a detector is also a concern. For example, consider the impact low-quality manipulations would have on moderation efforts in a form.  Since they are easy to identify, they would likely get flagged by these users rather than flagged by a manipulation detector, which would identify it as authentic.  Due to this conflict between the users and automatic detectors, a moderator may be tasked with reviewing the image as well, expending costly resources.
>
> What’s more, in Table 8 our work also highlights that manipulation detectors find low and high quality manipulations equally challenging, with little difference between the two sets. This illustrates an important observation made by our work: human judgements and machine detectors do not use similar evidence to identify manipulations.  This is to be expected to some degree, as automatic metrics judging image manipulation and generation quality are challenging to construct, making human judgements the gold standard for those tasks.  Thus, we show the quality of the manipulations has little to do with the goal of our work: creating high quality manipulation detection methods that generalize across a range of distribution shifts.
>
>
> >2) The image manipulation methods used in the paper are not very new. Using SD series, like SD2, or even SDXL is better.
>
> We emphasize that our focus in this paper is to broadly evaluate robustness of various image manipulation detection methods. Moreover, most prior datasets do not include any diffusion-based methods at all.  That said, we conducted an experiment whereby we inpainted 1000 images from both Magic-News and Magic-COCO using SDXL [j] and tested EVP and DOLOS on these images, additionally we have included the results from our prior experiments from Table 3 with our Stable-Diffusion results. Looking at the results overall we can see that testing on SDXL still showcases the similar problem we are highlighting, that OOD distribution performance on average performs worse than in-distribution performance.  In our camera ready we will expand on these initial results.
>
> |            |                | Stable-Diffusion-XL |                |                |
> |------------|----------------|---------------------|----------------|----------------|
> | Trained on | MAGIC-News     | MAGIC-COCO          | MAGIC-COCO     | MAGIC-News     |
> | Tested on  |   MAGIC-News   |      MAGIC-News     |   MAGIC-COCO   |   MAGIC-COCO   |
> |            | MT-OOD         | MT-OOD              | MT-OOD         | MT-OOD         |
> |            | AUC/F1/IoU     | AUC/F1/IoU          | AUC/F1/IoU     | AUC/F1/IoU     |
> | EVP        | 60.6/54.2/17.8 | 51.6/43.1/5.52      | 73.9/29.1/4.97 | 71.2/27.0/7.20 |
> | DOLOS      | 57.2/53.3/30.2 | 52.9/53.3/32.0      | 48.1/23.3/14.1 | 51.6/24.6/12.0 |
> |            |                | Stable-Diffusion    |                |                |
> | EVP        | 52.1/16.4/40.9 | 57.4/40.0/1.51      | 81.5/37.2/17.1 | 69.9/25.2/3.85 |
> | DOLOS      | 47.0/44.9/30.2 | 49.5/45.9/30.8      | 65.9/22.3/3.88 | 53.9/23.1/8.07 |
>
> [j] Podell, Dustin, et al. "Sdxl: Improving latent diffusion models for high-resolution image synthesis." arXiv preprint arXiv:2307.01952 (2023).

---

> ### Author Response · Authors · 2024-11-23
>
> >3) In Table 3, it's interesting that the OOD score is higher than the ID score when trained on MAGIC-News and tested on MAGIC-COCO. It's better to provide a more depth analysis.
>
> For the last two columns of Table 3, if we look at the instance of EVP trained on Magic-News and tested on Magic-COCO we can see that MT-ID is 62.1% and MT-OOD is 67.6% for AUC. We have provided a breakdown of the results for each of the inpaintings present for MT-ID and MT-OOD seen below. We see that for MT-OOD, EVP tends to perform well on Stable Diffusion with 69.9% and Blended Latent Diffusion 72.6% AUC versus MT-ID like GLIDE at 64.2% and Latent Diffusion at 67.9%. This suggests that the Stable Diffusion and Blended Latent diffusion images are easier to localize the manipulations when trained on Magic-News and tested on OOD. Because of the latent space that Stable Diffusion is generated from and because Blended Latent Diffusion combines Blended and Latent diffusion together it can help explain why there is a higher performance for these models.
>
>
> |        EVP        |                |                          |                |
> |:-----------------:|:--------------:|:------------------------:|:--------------:|
> | Trained on:       | Magic News     | Trained on:              | Magic News     |
> | Tested on:        | Magic COCO     | Tested on:               | Magic COCO     |
> |       MT-ID       |                |          MT-OOD          |                |
> | GLIDE             | 64.2/22.9/1.79 | Stable Diffusion         | 69.9/25.2/3.85 |
> | Latent Diffusion  | 67.9/24.4/3.49 | GLIGEN Splicing          | 57.8/23.5/2.70 |
> | Blended Diffusion | 71.5/26.5/3.58 | Blended Latent Diffusion | 72.6/27.6/8.25 |
> | Original          | 51.6           | Adobe Firefly            | 75.2/33.2/19.1 |
> | AUC Average       | 62.1           | AUC Average              | 67.6           |

---

> ### Author Response · Authors · 2024-12-01
>
> We would like to thank you for your review, as a reminder the discussion period is coming to an end. We have responded to your current questions, if you have any other questions or further clarifications to improve our score we would be happy to answer them.

---

### Official Review · Reviewer_LCuF · 2024-11-02

**Soundness:** 2
**Presentation:** 2
**Contribution:** 2
**Rating:** 5
**Confidence:** 3

**Summary:**

This paper introduces “MAGIC,” a large-scale dataset designed to evaluate the robustness and generalization of image manipulation detection models across multiple domains. MAGIC aims to assess model performance under various domain shifts, including different image sources, manipulation types, semantic topics, and manipulation scales. Results indicate that while the models perform well in distribution (ID), their OOD performance is limited, highlighting the challenges of domain generalization in image manipulation detection.

**Strengths:**

- MAGIC addresses a pressing issue in image manipulation detection by offering a large-scale dataset with a focus on domain generalization across multiple dimensions. This effort is commendable and fills a gap in manipulation detection research.

**Weaknesses:**

- The proposed dataset, while diverse, does not introduce fundamentally new manipulation detection methods or models. The dataset’s construction (e.g., sourcing from MS COCO and VisualNews, manipulation types) is novel but does not demonstrate significant methodological innovation beyond combining existing datasets and manipulation techniques. Thus, the contribution is more incremental than groundbreaking.

- The paper lacks a detailed comparison with other recent datasets or techniques, and the experiments primarily rely on existing architectures without substantial modifications or improvements. The work’s dependence on pre-existing models for analysis and lack of new methodological contributions weaken its overall technical impact.

I am not an expert in this field. But I think that a dataset for image manipulation localization is not sufficient for publication at ICLR.

**Questions:**

See weakness.

---

> ### Author Response · Authors · 2024-11-23
>
> We thank the reviewer for spending the time on this paper and giving us feedback, please see our replies below.
>
> >The proposed dataset, while diverse, does not introduce fundamentally new manipulation detection methods or models … The proposed dataset only combines existing datasets and manipulation techniques …Thus, the contribution is more incremental than groundbreaking.
>
> For our proposed dataset our main focus was not to propose a new method for inpainting or new models for detecting inpaintings. But to produce a new diffusion based inpainting benchmark dataset that allows researchers to study generalization across several axes of generalization for image manipulation detection. A major drawback of current image manipulation datasets is that they typically only focus on one axis of generalization, namely manipulation type and do not allow researchers the ability to determine how their models perform under axes like image source as well as manipulation type. In our experiments we highlight a major disadvantage of current models being able to generalize across these different axes as shown in Table 3 and highlight the need for better generalization for these image manipulation detection models
>
>
> >The paper lacks a detailed comparison with other recent datasets or techniques … The experiments only rely on existing architectures without substantial modifications.
>
> In our paper we can see that in Table 1 we compare our model to other current manipulation detection datasets, additionally we discuss these datasets in the Related works section (Section 2). We additionally did provide experiments utilizing two domain generalization techniques namely Model Soups and SWAD and showed that these off the shelf techniques do not significantly improve performance in terms of generalization. We will highlight that this is a benchmark paper designed to highlight issues with existing methods, hence our main focus is not on providing new techniques in this paper. As you can see below [c,d,e], we have a number of papers being published at ICLR that do not propose new methods.
>
> As this is an topic with increasing importance as image maniuplation methods become more sophisticated, our task's importance is likely only to grow.  This makes datasets like ours that explore the generalization capabilities of these models and is complementary to existing datasets of vital importance for defending against misinformation such as applications to fighting crime (as defendants may claim an image has been altered by an AI model) and content moderation (where manipulated images may be presented as real).  These are high impact applications that are meant to increase safety and security, highlighting their importance for study in our paper.
>
> [c] Wang, Xingyao, et al. "Mint: Evaluating llms in multi-turn interaction with tools and language feedback." ICLR 2024
>
> [d] Gu, Jiuxiang, et al. "ADOPD: A Large-Scale Document Page Decomposition Dataset." ICLR 2024
>
> [e] Wu, Haoning, et al. "Q-bench: A benchmark for general-purpose foundation models on low-level vision." ICLR 2024

---

> ### Author Response · Authors · 2024-12-01
>
> We would like to thank you for your review, as a reminder the discussion period is coming to an end. We have responded to your current questions, if you have any further questions or clarifications to improve our score we would be happy to answer them.

---

### Official Review · Reviewer_iuZd · 2024-11-03

**Soundness:** 3
**Presentation:** 2
**Contribution:** 3
**Rating:** 6
**Confidence:** 3

**Summary:**

The paper primarily builds a Diffusion-based model and constructs an image manipulation localization dataset with 192k images focused on inpainting tampering types. This dataset is divided into three major categories based on source, content topic, and specific types of Diffusion models used, allowing for evaluation of the model's cross-domain generalization performance. The authors evaluated the performance of several SoTA models on this dataset. A survey on human feedback on the quality of this dataset is also reported.

**Strengths:**

- The motivation is sound, as cross-dataset or cross-domain performance has consistently posed challenges in the field of image manipulation localization. A dataset focused on cross-domain performance analysis would serve as a valuable benchmark.
- Experiments are comprehensive in demonstrating the utilization of each protocol.

**Weaknesses:**

## Main issue
- Although the authors claim to have used clustering for categorizing topics, some topics displayed in Figure 2 seem to vary significantly and do not appear entirely reasonable. For instance, it’s unclear how a bicycle image relates to the "ARTS" category, and both "People" and "Ruins" are grouped under "Media." Additionally, the results across many topic classes in Table 4 are quite similar, suggesting that the distribution between classes may not be as distinct as initially anticipated.
- A paper [A] proposed in Nov. 2023 on ArXiv and accepted by ACM MM 2024 also uses Visual News and COCO to create a fine-grained diffusion and GAN-generated dataset. I understand that ACM MM was held after ICLR submission. However, the two articles have considerable similarities in the background, purpose, and subject matter. While the two papers represent distinct works, it is recommended to discuss [A] in the Related Work section and reconsider the claim in line 149 regarding being the "first diffusion-based manipulation dataset." Since [A] retains the text prompts associated with images, which, with proper handling, could serve as a more accurate basis for topic categorization.
- For dataset-focused papers, it’s common to include tests with standard vision backbones, such as ResNet or Swin, to provide more straightforward benchmarks. Including these could serve as helpful references for comparison.

## Minor issue
- Many AUC metrics in the tables are too close, showing little distinction and some values are excessively low. For instance, Table 3 includes numerous metrics below 0.5, indicating that the model has not effectively learned the corresponding distributions. More distinctive metrics, such as F1 or IoU, may be needed better to assess the model’s performance on each protocol.

- The statement between lines 522–524 appears unconvincing. The current explanation does little to clarify why the performance of PSCC is nearly the opposite of the other two models.


## Reference
[A] Zhihao Sun, Haipeng Fang, Juan Cao, Xinying Zhao, and Danding Wang. 2024. Rethinking Image Editing Detection in the Era of Generative AI Revolution. In Proceedings of the 32nd ACM International Conference on Multimedia (MM '24). Association for Computing Machinery, New York, NY, USA, 3538–3547. https://doi.org/10.1145/3664647.3681445

**Questions:**

See the Weakness Section. Overall, this is a solid piece of work. I will consider raising my rating if improve the presentation of details.

---

> ### Author Response · Authors · 2024-11-23
>
> We thank the reviewer for spending the time on this paper and giving us feedback, please see our replies below.
>
> >Although the authors claim to have used clustering for categorizing topics, some topics displayed in Figure 2 seem to vary significantly and do not appear entirely reasonable. For instance, it’s unclear how a bicycle image relates to the "ARTS" category, and both "People" and "Ruins" are grouped under "Media."
>
> The grouping of the images is not based on image content, but rather the article content.  Thus, a bicycle could fall under arts due to an event with bicycles, or even simply a photography competition.  In addition, these categories are broad, and contain many subcategories (i.e., the original VisualNews dataset had 159 subcategories that represent distinct subsets of our 8 categories).
>
> >The results across many topic classes in Table 4 are quite similar, suggesting that the distribution between classes may not be as distinct as initially anticipated
> There is nearly a 7 point difference between the best and worst performing topic within Table 4.  While this is not as large as some other shifts, it still provides a significant difference to compare various models.
>
> >A paper [A] proposed in Nov. 2023 on ArXiv and accepted by ACM MM 2024 also uses Visual News and COCO to create a fine-grained diffusion and GAN-generated dataset … it is recommended to reconsider the claim “"first diffusion-based manipulation dataset." on line 149.
>
>
> Thank you for pointing out this paper and we have adjusted our associated discussions.  We are unable to add the information to Table 1 at this time as many of the detailed statistics that breaks down the type of manipulation are not reported and we could not get access to the dataset in time for this response (although we have requested access and will include this information in our paper once we have been granted access). To summarize our comparisons, our dataset has many complementary benefits, as there is little overlap in the type of manipulation techniques utilized between our datasets.  In addition, we focus on the impact of several axes of generalization that [A] did not study, including studying distribution shifts due to image statistics or topics.  We also study the effect of manipulation quality on detection performance, whereas [A] did not consider this component.  As such, our paper provides several notable contributions over even concurrent work like [A].
>
> >Since [A] retains the text prompts associated with images, which, with proper handling, could serve as a more accurate basis for topic categorization.
>
> As we noted earlier, our topics are based on article content as opposed to image content.  As these prompts are associated with image content, they are more akin to COCO captions, which our dataset also has access to (for the COCO subset).  We also argue that comparing VisualNews vs. COCO images already provides an example of changes due to image content.

---

> > ### Comment · Reviewer_iuZd · 2024-11-25
> >
> > Thanks for your response,
> >
> > I think the answer to the first issue about the category is not very satisfying. At least, this will somewhat diminish the contribution of the paper. If the content itself does not align with its category but needs the support of description, then it's hard to declare as "across domains" reasonable. It's as if the boundaries between the two domains are not very clear, with possible overlapping areas, which weakens this claim. The experiment result decreases for multi-domain may be coming from overfitting instead of OOD quality.
> >
> >
> > The discussion about [A] is clear and has already been added to the manuscript.

---

> > > ### Author Response · Authors · 2024-11-26
> > >
> > > >I think the answer to the first issue about the category is not very satisfying. At least, this will somewhat diminish the contribution of the paper. If the content itself does not align with its category but needs the support of description, then it's hard to declare as "across domains" reasonable. It's as if the boundaries between the two domains are not very clear, with possible overlapping areas, which weakens this claim. The experiment result decreases for multi-domain may be coming from overfitting instead of OOD quality.
> > >
> > > Thank you for participating in the discussion of our paper! While these categories are based on the article content, the images also have statistical differences between them.  For example, in MAGIC-New’s Politics and Elections category a person was the manipulated object in nearly half the images as this topic is far more person-centric than say, Science Technology, where they were selected in less than a quarter.  A “rider” exists in 4-8x more images in Sports and Entertainment than other categories.  Thus, there are important differences between the images contained within these topics, so it is important to know if a model can generalize.
> > >
> > > That said, these broad categories are also quite similar to how these images may be used in practice, as the topics we used are based on those produced by the news websites.  However, these types of loose groupings are not limited to news. For example, a subreddit within the website Reddit can be focused on a particular board theme, which would be analogous to our topics.  Images that post there would also likely have statistical differences with images of other subreddits, even if they can be hard to understand the relationship out-of-context of the post they stem from.

---

> > > > ### Comment · Reviewer_iuZd · 2024-11-26
> > > >
> > > > I see the point. Discussing the gap from the distribution of object semantics is more convincing than simply talking about the label.
> > > >
> > > > I think more analysis on the distribution gap of semantics should be included in the paper, even intuitively, to defend and support your arguments of cross-domain (at least in supplementary material if there exist space limitations).
> > > >
> > > > Anyway, most of my doubts are solved, I have raised my rating from 5 to 6.

---

> > > > > ### Author Response · Authors · 2024-12-01
> > > > >
> > > > > We would like to thank you for putting the time and effort to participate in the discussion period for our paper and we thank you for raising your score. We know that the period to revise our pdf has passed, however we will add our experiments and analysis for the camera ready once our paper is accepted.

---

> ### Author Response · Authors · 2024-11-23
>
> >For dataset-focused papers, it’s common to include tests with standard vision backbones, such as ResNet or Swin … as a standard baseline.
>
> >Many AUC metrics in the tables are too close … Table 3 includes numerous metrics below 0.5, indicating the models have not effectively learned the corresponding distributions. More distinctive metrics such as F1 or IoU, may be needed to assess the model’s performance on each protocol.
>
> We trained a Swin Transformer [g] trained to make pixel-level predictions using the Upernet strategy [h] for manipulation localization to act as a standard baseline. The results are shown below and we have added these results to Table 3.  As shown below, most specialized models perform worse than a simple Swin-based model.
>
> We have also added the F1 scores to Table 3, partly reproduced below with IoU scores as well, as you suggested. We observed that the F1 and IoU scores are also generally low across both manipulation OOD and image source OOD settings. The observed scores, particularly the low AUC (along with F1 and IoU) values in certain cases, reflect the core focus of our paper: highlighting that current models struggle to generalize across different domains for the problem of image manipulation detection.
>
> | Trained on | MAGIC-News     |                | MAGIC-COCO     |                | MAGIC-COCO     |                | MAGIC-News     |                |
> |------------|----------------|----------------|----------------|----------------|----------------|----------------|----------------|----------------|
> | Tested on  |                |   MAGIC-News   |                |                |                | MAGIC-COCO     |                |                |
> |            | MT-ID          | MT-OOD         | MT-ID          | MT-OOD         | MT-ID          | MT-OOD         | MT-ID          | MT-OOD         |
> |            | AUC/F1/IoU     | AUC/F1/IoU     | AUC/F1/IoU     | AUC/F1/IoU     | AUC/F1/IoU     | AUC/F1/IoU     | AUC/F1/IoU     | AUC/F1/IoU     |
> | EVP        | 79.2/73.4/61.6 | 61.9/49.0/26.7 | 62.0/49.2/14.7 | 55.7/44.9/3.39 | 79.0/50.7/35.8 | 79.6/38.4/20.6 | 62.1/23.9/2.66 | 67.6/25.2/3.85 |
> | EVP + SWAD | 79.5/73.2/55.7 | 62.8/53.4/22.9 | 60.2/45.2/11.7 | 56.7/39.4/1.77 | 79.2/52.8/37.2 | 84.3/42.3/22.6 | 58.3/22.5/1.53 | 66.9/24.3/3.27 |
> | EVP + Soup | 80.9/74.9/63.8 | 64.2/51.8/29.2 | 63.7/47.7/17.4 | 57.4/33.6/3.63 | 80.6/57.8/43.4 | 84.0/43.5/23.1 | 54.9/20.8/3.77 | 59.8/22.0/1.25 |
> | DOLOS      | 78.1/71.1/61.4 | 57.0/49.0/34.4 | 69.6/55.4/36.6 | 59.7/52.4/38.2 | 61.3/21.8/2.59 | 62.0/23.3/6.63 | 48.5/21.8/3.39 | 52.9/24.3/9.39 |
> | PSCC-Net   | 72.9/72.8/61.6 | 49.5/48.9/37.5 | 51.8/29.0/8.91 | 49.7/3.90/0.44 | 71.6/36.8/23.6 | 70.2/30.9/17.5 | 48.8/4.78/0.30 | 49.7/4.50/0.32 |
> | HiFi       | 73.6/77.8/70.9 | 50.9/29.9/21.4 | 49.6/10.1/6.45 | 48.6/1.85/1.09 | 66.8/34.6/24.0 | 62.7/21.5/14.2 | 51.5/5.73/3.71 | 51.6/7.55/4.51 |
> | Swin   | 65.9/63.8/56.8 | 57.0/44.5/36.0 | 54.3/21.6/16.3 | 50.1/0.91/0.63 | 60.6/31.6/24.6 | 57.3/15.5/11.3 | 50.1/0.70/0.53 | 49.9/0.04/0.02 |
>
>
> [g] Liu, Ze, et al. "Swin transformer: Hierarchical vision transformer using shifted windows." Proceedings of the IEEE/CVF international conference on computer vision. 2021.
>
> [h] Xiao, Tete, et al. "Unified perceptual parsing for scene understanding." Proceedings of the European conference on computer vision (ECCV). 2018.

---

> > ### Comment · Reviewer_iuZd · 2024-11-25
> > **Clear experiments on AUC and IOU**
> >
> > The supporting experiments on AUC and IOU provide a clearer visualization of the model's performance. In many cases, if the F1 score is lower than 0.1, the performance can be considered worse than simply predicting an entirely white output.
> >
> > The conclusion is clear that IML-oriented methods perform better on OOD datasets compared to common vision backbones. Further, both of them struggle with OOD detection. (Although I still have a little doubt on if the OOD exactly represent crosses the domain)

---

### Author Response · Authors · 2024-11-24

We would like to thank all the reviewers for their feedback. We are encouraged that they recognize the sound motivation behind our work (*Reviewer iuZd*) addressing a pressing issue (*Reviewer LCuF*); they comment on the benchmark as being valuable (Reviewer iuZd), good (*Reviewer GsP7*), covering many scenarios, editing operations/techniques (*Reviewer VNC9*); they state that our experiments are comprehensive (*Reviewer iuZd*) and good (*Reviewer GsP7*), and an overall effort is commendable (*Reviewer LCuF*).

Reviewers comments included:

1) Questions as to how the quality of the manipulations affect our task (*Reviewers GsP7, VNC9*), which we pointed to results in Table 8 where image maniuplation localization models found high and low quality maniuplation samples similarly challenging.

2) Request for additional metrics like F1 score or IoU (*Reviewers iuZd, VNC9*), which we have included in Table 3 and are consistent with the metrics we already reported.

3) Request for a baseline Swin Transformer approach (*Reviewer iuZd*), which we have included in Table 3.

4) Request for results on SDXL (*Reviewer GsP7*), which we provided with some initial results that has similar behavior to other generators.

5) Various presentation changes that have created or modified the figures and writing in our paper (directly responded to for each reviewer)

We are happy to make any additional adjustments to our paper suggested by reviewers.

---

### Meta-Review · Area_Chair_tytE · 2024-12-20

**Metareview:**

This paper introduces a new dataset designed to evaluate the robustness and generalization of image manipulation detection models across multiple domains. Initially, the reviewers raised several issues. That includes the quality of the constructed dataset (both image quality and the quality of the clustered topics), the lack of new manipulation detection methods, and the image manipulation methods used in this paper are not up-to-date. After rebuttal, most of the concerns are addressed. Also, as a dataset-focused paper, proposing new methods is not inherently required. Yet the reviewers remained significantly concerned about the low image quality within the proposed dataset, which is considered to be crucial given its intended role as a benchmark for image manipulation detection. For the dataset to be more valuable to the community, its image quality should be further improved. Overall, this paper is considered borderline. While it offers some valuable contributions, its current state does not fully meet the criteria for clear acceptance. To strengthen its position, the authors should enhance the dataset's quality to establish a more robust evaluation benchmark.

**Additional Comments On Reviewer Discussion:**

After rebuttal, the reviewers still share concerns about the image quality of the proposed dataset.

---

### Decision · Program_Chairs · 2025-01-22

Reject